# Astrocytes in the Neuropathology of Bipolar Disorder: Review of Current Evidence

**DOI:** 10.3390/brainsci12111513

**Published:** 2022-11-08

**Authors:** Nasia Dai, Brett D. M. Jones, Muhammad Ishrat Husain

**Affiliations:** 1Department of Pharmacology and Toxicology, University of Toronto, Toronto, ON M5S 1A8, Canada; 2Department of Psychiatry, Temerty Faculty of Medicine, University of Toronto, Toronto, ON M5T 1R8, Canada; 3Campbell Family Mental Health Research Institute, Centre for Addiction and Mental Health, Toronto, ON M5G 2C1, Canada

**Keywords:** bipolar disorder, neuroinflammation, astrocytes, astroglial activation, astrocyte dysfunction

## Abstract

(1) Background: Approximately one-third of patients with bipolar disorder (BD) do not experience sustained remission with current treatments. Presently, astrocytes, i.e., glial cells that act as key regulators of neuroinflammation, have been a target for therapeutic development. Research regarding their role in the neuropathology of BD is limited. We conducted a scoping review on evidence linking astrocytes to the pathology of BD. (2) Methods: The search was conducted in MEDLINE for studies published from inception to August 2022. Studies of interest were data-extracted and reported based on the Preferred Reporting Items for Systematic Reviews and Meta-analysis Protocols. (3) Results: Overall, 650 publications were identified, of which 122 full texts were evaluated and 12 included. Four were in vitro, seven were ex vivo, and one study was both in vitro and in vivo. In vitro investigations focused on plasma levels of neuroinflammatory biomarkers S100B and glial fibrillary acidic protein (GFAP). Ex vivo investigations were post-mortem brain studies assessing astrocytes in regions of interest (i.e., anterior cingulate cortex, dorsolateral prefrontal cortex) using phosphorylated GFAP and ASCT-1. The in vivo and in vitro study evaluated morphological and chemical variations of YKL-40 between cohorts. (4) Conclusions: Reports indicate an association between astrocyte dysfunction and BD although larger studies are required to validate this association.

## 1. Introduction

Globally, 40 million people are diagnosed with bipolar disorder (BD) [1,2,3,4,5]. BD is a prevalent and severe mood disorder in which patients exhibit a combination of recurrent major depressive episodes in addition to hypomanic or manic episodes [5]. Due to its chronic and recurrent nature, individuals with BD experience increased morbidity and mortality [6]. There is also a substantial burden related to suicidal ideation and attempted and completed suicide [4,7]. Though current treatments (i.e., mood stabilizers, antipsychotics) are effective, up to one-third of individuals with BD do not achieve sustained remission with currently available treatments [8,9]. A major contribution to the relative treatment resistance is that none of these medications were developed to target disease-specific pathology. To improve outcomes for those with BD through targeted novel treatments, a greater understanding of the pathophysiology of BD is required [4,10].

There has been increasing interest in the role of the immune system in the pathophysiology of BD. High rates of inflammatory comorbidities such as autoimmune disorders and cardiovascular disease have been linked to BD, and pro-inflammatory processes such as pathological glial over-activation and increased oxidative stress have been reported in individuals with BD [11]. Recent work has implicated environmental factors in the activation of the inflammatory response system in BD. For instance, it has been postulated that the mechanism by which mania/hypomania manifests in a subgroup of individuals is through a dysregulated internal clock and sleep disturbances. Patients with BD have been reported to experience irregular sleep-wake rhythms, resulting in abnormal melatonin secretion and low-grade inflammation. Disruptions in the circadian rhythms indicate that environmental factors may play a contributory role in neuroinflammation and disease manifestation of BD [12].

Given converging evidence on associations between an activated inflammatory response and BD, several studies investigating immune modulating drugs for the treatment of BD have been published albeit with conflicting results [13]. A major critique of published clinical trials has been the lack of a stratified approach to target specific subgroups of patients with BD that display evidence of neuroimmune dysfunction.

Astrocytes, a class of glial cell, are key cells in the central nervous system (CNS) that play a role in neuronal maintenance and regulation [14]. Astrocytes maintain CNS homeostasis during development, normal physiology, and aging [15]. As an integral part of normal brain milieu, these cells provide trophic support to neurons, mediate synapse formation and function, prune synapses during development, and assist in signaling across brain regions to modulate neuronal activity [15]. Astrocytes have an important role in the blood–brain barrier (BBB) to sense and respond to neuroinflammation and transport metabolites to fuel the brain [15]. In response to brain injury, astrocytes are “activated” and undergo a morphological process termed astrogliosis. This serves as a defense mechanism to provide neuroprotection and regeneration after the initial damage [16,17]. Disturbances in this process as evidenced by increased or reduced astrogliosis have been implicated in CNS pathology [18].

Studies have reported alterations in astrocyte morphology, function, and abundance in neuropsychiatric disorders as leading to disturbances in mood symptoms and suicidality [18]. Specifically, in BD, reduction in glial cell numbers, particularly astrocytes, have been reported [19]. Deficiency in astroglial cell density and function may therefore contribute to the disease progression of BD [19]. Though prior research has found increased levels of microglial biomarkers in BD patients, astrocytes are the most abundant glial cell in the CNS and may play a key role in the pathology of neuropsychiatric disorders, including BD [20,21,22]. We conducted a scoping review to synthesize evidence linking astrocyte activity to BD to inform the development of further research that aims to understand the pathophysiology of this prevalent and disabling mental disorder.

## 2. Materials and Methods

The protocol for this study was drafted using the Preferred Reporting Items for Systematic Reviews and Meta-analysis Protocols (PRISMA-P) [23] Specifically, the scoping review was conducted using the PRISMA extension for scoping reviews published in 2018 [23].

The search was conducted in MEDLINE for studies published from inception to August 2022. The following keywords were used to identify publications of interest: “Bipolar Disorder”, “Bipolar Depression”, “Manic Depression”, “Bipolar Affective Disorder”, “Astroglial”, “Astrocytes”, “Astrogliosis”, “Glial”, and “Neuroinflammation”. Eligible studies were those in English; involving human participants with a DSM-5 diagnosis of BD I or BD II; and examining astrocyte activation (i.e., astrogliosis) in vitro, in vivo, or ex vivo. Studies were excluded if they did not meet the eligibility requirements listed previously. Studies of interest were data-extracted, and quality was assessed.

One author (N.D.) evaluated the titles, abstracts, and then the full text of all publications identified by the search. Disagreements on study selection and data extraction were resolved by consensus and discussion with two co-authors (B.D.M.J. and M.I.H.).

A data-charting form was developed by two reviewers (N.D. and B.D.M.J.) to evaluate which variables to extract. Independently, data were charted, and results were discussed to continuously update the data-charting form in an iterative process. Data on article characteristics (e.g., publication year, type of study conducted), study focus (e.g., mood disorder evaluated), contextual factors (e.g., diagnostic tools, statistics), limitations, and results of any formal assessment of engagement were reviewed. The search was completed on 29 August 2022.

## 3. Results

A total of 650 publications were identified, of which 122 full texts were evaluated (Figure 1). In the present scoping review, 12 were included, with 4 studies being in vitro, 7 ex vivo, and 1 study being both in vitro and in vivo.

### 3.1. In Vitro Studies

Identified in vitro studies consisted of peripheral markers that were typically collected from serum or whole blood (Table 1). The most common peripheral markers of astrocyte activation have been S100B and glial fibrillary acidic protein (GFAP).

#### 3.1.1. S100B

S100B is a calcium-binding protein derived from astrocytes that affect neurons and glia in the CNS [24,26]. Increased extracellular levels of the protein leads to reactive oxygen species (ROS)-induced stress, cell death, and toxicity, resulting in brain damage and neurodegeneration [24,26]. Neurodegeneration is associated with elevated S100B levels in relation to astrocytic death, astrogliosis, or active secretion for neuronal damage repair [24].

One study evaluated brain injury and astrocyte activity using serum S100B in BD patients during different phases of illness [24]. This was a case-control study of 84 adults with BD (*n* = 21 depressed, *n* = 32 manic, *n* = 31 euthymic, *n* = 32 health). There was significant increase in S100B in depressive (*p* = 0.004) and manic (*p* = 0.011) BD phases compared to controls. There were no differences identified in euthymic patients (*p* = 0.263) compared to healthy controls (HC) [24]. All subjects received medication, and two-thirds received lithium. Another study quantified serum S100B levels in 20 unmedicated BD adults with mania vs. 20 HC. Consistently, there were significantly higher levels of S100B in the subjects with mania compared to controls (Wilcoxon signed-ranks test, Z = −2.45, *p* = 0.01) [26]. In another study, treatment of mania (conducted for 27.7 ± 4.8 days) resulted in a significant decrease in serum S100B levels in adults with BD (*n* = 17) although baseline levels of S100B did not differ compared to HC [27].

#### 3.1.2. Glial Fibrillary Acidic Protein

Glial fibrillary acidic protein (GFAP) is the prominent intermediate filament protein in astrocytes. During processes such as CNS trauma, ischemia, and neurodegeneration, astrocytes assume a reactive phenotype (i.e., astrogliosis), which is marked by the upregulation of GFAP [28]. We identified one study of GFAP levels in BD investigating the effect of long-term lithium treatment on glial markers (including GFAP) [29]. The study utilized real-time quantitative reverse transcription PCR to identify mRNA levels of glial markers (GFAP, Olig1, and Olig2) from venous blood samples. The study consisted of 30 adults with BD in remission, divided into two groups: 15 treated with lithium and 15 lithium-naïve individuals. There was a significantly higher mean GFAP mRNA levels in the lithium naïve group compared to the lithium-treated group [29]. There was no significant difference in GFAP mRNA levels between the BD lithium-treated group and HC [29].

### 3.2. Ex Vivo Studies

Ex vivo studies identified were predominantly post-mortem brain studies assessing neuronal and glial cells in regions of interest (Table 2). All seven studies identified utilized brain specimens from the Stanley Foundation Brain Consortium or Stanley Medical Research Institute. Only one of the studies focused on BD, two evaluated BD and schizophrenia (SCZ), and four investigated BD, SCZ, and major depressive disorder (MDD) [30,31,32,33,34,35,36].

One post-mortem study aimed to determine the location of phosphorylated isoforms of GFAP (pGFAP) using mono-clonal antibodies in the pre-frontal cortex and hippocampus [35]. The study compared levels of GFAP expression between subjects with schizophrenia (*n* = 15), BD (*n* = 15), MDD (*n* = 15), and HC (*n* = 15). Immunohistochemical localization of pGFAP distribution in the prefrontal cortex (PFC) and hippocampus identified astrocytes that were distributed in patches of various sizes and intensity along the pial surface [35]. Astrocytes positive for pGFAP were also found to be adjacent to blood vessels in the white matter of the gyrus and the polymorphic layer of the dentate gyrus of MDD, SCZ, and BD groups [35]. There were no significant differences between the BD group and control group [35].

Another post-mortem study of subjects with BD (*n* = 10), SCZ (*n* = 9), MDD (*n* = 11), and HC (*n* = 14) found a reduction in the density of GFAP immunoreactive astrocytes in the amygdala of the MDD group compared to all other groups; however, there was no difference between SCZ, BD, and HC [36]. In another study utilizing 15 participants per group from the same consortium, there was significant reduction of GFAP mRNA in the anterior cingulate cortex (ACC) white matter in the SCZ group and BD group compared to controls [34].

Another study examined oligodendrocyte, astrocyte, and microglial populations in post-mortem white matter of subjects with BD and schizophrenia compared to HC [31]. The area fraction of GFAP (*p* = 0.05) and astrocyte spatial distribution (*p* = 0.040) significantly differed between groups [31]. BD samples exhibited decreased GFAP area fractions and increased cell clustering [31]. Astrocyte density did not differ between psychotic groups, respectively [31].

Given inconsistencies in reports of GFAP, studies have explored various other markers in post-mortem tissue. For instance, one study evaluated multiple astrocyte-specific proteins associated with the activated and non-activated state: GFAP, aldehyde dehydrogenase 1L1 (ALDH1L1), vimentin, and excitatory amino acid transporter 1 (EAAT1) [30]. The sample included HC (*n* = 35), BD (*n* = 34), and SCZ (*n* = 35). Elevated levels of GFAP, aldehyde ALDH1L1, vimentin, and EAAT1 were reported in BD [30]. Elevated GFAP levels were primarily observed in BD samples that exhibited psychotic symptoms compared to those without [30].

Another post-mortem study investigated if the pattern and cellular localization of ASCT-1 protein varied among BD, MDD, and SCZ samples [33]. In the hippocampus, there was an immense loss of immunoreactivity on astrocytes, neurons, and interneurons in multiple regions in BD samples compared to MDD and HC, respectively [33].

Another study investigated sex differences of glia gene expression in the dorsolateral prefrontal cortex (DLPFC) in BD (*n* = 30) [32]. The authors utilized real-time polymerase chain reaction (PCR) to detect transcriptional alterations of 16 glia-related genes in two brain areas, the DLPFC and ACC [32]. Samples from subjects with BD were subdivided by suicide and psychotic features and matched to 34 control cases [32]. Associations between transcriptional changes of astrocyte, microglia, and oligodendrocyte markers with suicide and psychotic features were analyzed to identify any abnormalities in the DLPFC and ACC [32]. Sex differences of glia-related gene expression were found to be significant in the DLPFC of BD subjects [32]. Males exhibited higher expression of the genes, respectively. Fourteen glia-related genes were expressed significantly higher in males, including microglial and astrocyte cells [32].

### 3.3. In Vivo and In Vitro Study

A 2022 study that utilized in vitro and in vivo experiments investigated the chemokine YKL-40, which undergoes transcription in astrocytes [37]. YKL-40 is associated with cell migration and morphological changes that are an indicator of reactive gliosis [37]. The study recruited 31 euthymic bipolar subjects aged 20 to 45 [37]. Using blood samples and MRI scans, the study reported that significantly higher YKL-40 and sTNF-R1 levels were associated with lower volumes of the left anterior cingulum, left frontal lobe, right superior temporal gyrus, and supramarginal gyrus [37] (Table 3).

## 4. Discussion

The current scoping review synthesized evidence linking astrocyte dysfunction to BD. The review identified twelve studies examining surrogate markers of astrocyte activity in both in vivo and ex vivo samples. Four studies evaluating peripheral markers (S100B and GFAP) of astrocyte function in BD suggest that astrocyte function may be associated with mood states in BD; however, the small sample sizes of current studies limit clinical inference and require validation in larger samples that evaluate specific mood states.

Post-mortem and in vivo imaging studies could provide direct evidence of central disturbances in astrocyte function. We synthesized evidence from seven studies that investigated astrocyte function in post-mortem BD brains. There were conflicting results, with some studies reporting no difference in markers of astrocyte activity and some showing elevated activity and, in some cases, showing reduction of activity. These studies are also limited by the small samples, and further in vivo studies, particularly sophisticated neuroimaging approaches, are encouraged.

From all the studies identified, there was no consistency in terms of a specific peripheral biomarker to evaluate astrocyte activity. Previously, transcriptome studies have been conducted to investigate astrocytic markers. Astrocytic gene S100B was found to vary based on developmental stages. Elevated or reduced expression of the gene response may therefore reflect astrocyte function [38]. Marker gene profile (MGP) is a technique that can be used to assess various specific cell types in select samples. Tolker et al. (2018) reported high expression of MGP in region-specific astrocyte marker genes and prevalent cortical astrocyte marker gene. The clear transcriptomic signal insists cell type-specific changes in BD, and thus, investigations on which astrocyte biomarkers are prevalent and their respective regions of the brain should be assessed [39].

Recently, there has been growing interest in the potential of extracellular vesicles as putative peripheral biomarkers for a range of neuropsychiatric disorders [40]. Present studies regarding BD have not specified a cell type, neurotransmitter, or brain region that results in the neuropsychiatric disorder. Alterations in exosome levels are now an area of interest, as they are heavily involved in transport of proteins between cells in the nervous system and cell–cell communication [41]. Exosomes are a class of extracellular vesicles, which are released by neurons and other cells (including astrocytes) in the CNS and are accessible in the periphery through examination of saliva, urine, and plasma [42]. Exosomes have been shown to modulate synaptic plasticity, neuronal stress response, cell-to-cell communication, and neurogenesis [40]. Astrocyte-derived exosomes may be pragmatic peripheral markers of astrocyte function, as astrocytes regulate synaptic cleft homeostasis, and cargo is dependent on mechanical or inflammatory insults [19,43]. We are unaware of any studies assessing astrocyte-derived exosomes in individuals with BD, and further assessment of these cargo proteins may provide further evidence on the role of astrocyte function in the pathophysiology of BD.

Prior post-mortem research on astrocytes and oligodendrocyte numbers in BD were strongly associated with glial reduction [44] Morphometric alterations in glia and pyramidal neurons in the ACC have indicated aberrant functioning and connectivity, which may be associated with abnormal glial cell function in mood disorders [45]. Specifically, in BD patient groups, reduced levels of isoforms of GFAP in the frontal lobes and S100B-immunocontent of astrocytes in the hippocampus have been reported [46,47,48]. These findings were challenged by other studies that reported no differences in GFAP levels in the PFC between BD samples and HCs and no differences in astrocyte density in the amygdala of BD groups [45,49]. The influence of astrocyte function in BD is yet to be established, and further research is required, focusing on addressing the substantial heterogeneity in BD samples.

Lithium has established neuroprotective effects that may be in part due to its actions on astrocytes. Lithium’s neuroprotective effects may be pleiotropic through several actions, including modulation of nerve growth factors and reduction of inflammation, mitochondrial function, oxidative stress, and apoptosis or autophagy [50]. Pre-clinical studies have demonstrated that astrocytes may be a direct target of lithium treatment, demonstrating the potential of a new drug target for mood disorders [25,51]. Lithium use in mice resulted in astrocyte morphological and genomic phenotypic changes, which increased peroxisome proliferator-activated receptor gamma (PPAR-γ), a lithium-responsive pathway [49]. Pioglitazone, a drug specific for this pathway, may be a potential astrocyte-modifying BD novel therapeutic based on its antidepressant activity in preliminary clinical trials [52,53]. Further studies in clinical populations assessing associations between astrocyte function and lithium response may help provide a greater understanding on lithium’s mechanism of mood-stabilizing action and hence the pathophysiology of BD.

Preliminary evidence suggests that astrocyte dysfunction could be a target of BD treatment development given the cytotoxic and cytoprotective effects of these neuronal cells [54]. There are multiple peripheral and central markers of astrocyte function; however, only a small number have been investigated in BD thus far [54]. Peripheral markers of astrocyte function are advantageous, as they are relatively inexpensive and could be routinely used in clinical practice. Central markers of astrocyte function are either expensive and challenging to quantify (e.g., through PET imaging) or rely on post-mortem sampling. Studies in MDD have explored correlations between putative central markers of astrogliosis (e.g., monoamine oxidase B; MAO-B) with more common peripheral surrogate markers (GFAP) [55]. Further research is needed to evaluate the importance and expression of other peripheral astrocyte biomarkers such as aldehyde dehydrogenase family 1 member L1 (Aldh1L1), aldolase C (AldoC), glutamate transporter-1 (Glt1), and Aquaporin 4. PET imaging presently relies on using ^11^C-SL25.1188, a reversible MAO-B radiotracer, as well as 11C-PK11195 and 18F-DPA-714, which are radiotracers that target translocator proteins in astrocytes. Utilizing these radiotracers in PET imaging in conjunction with peripheral GFAP assessments in subjects with BD may allow for evaluation of reliable central markers of astrocyte function across mood states.

It is important to continue to evaluate astrocytes in BD in relation to neuroinflammation. Neuroinflammation has been found to result in astrogliosis. Astrocyte loss may be due to cytotoxic T cells, IL-1, and IL-6 [56]. Some BD studies have evaluated immune cells as markers of interest, yet there are limited data on associations between peripheral inflammatory markers and astrocytes. A study of patients with schizophrenia reported that IL-1B exposure led to decreased chemotactic effect on T cells, suggesting astrocyte-derived chemokine CCL20 may be the cause of neuroinflammation in psychotic illness [57]. Given the potential pathophysiological overlap between schizophrenia and BD, similar approaches should be considered in patients with BD.

The current scoping review had some limitations. We conducted our search using only one database (MEDLINE), thus potentially missing some studies in other databases. The poor quality of the studies included was also a limitation. Most did not report the mood episode of included subjects (i.e., manic, hypomanic, etc.). This limits our knowledge on whether astroglial dysfunction is “state”- or “trait”-dependent. An understanding on whether specific mood states affect astrocyte function may provide insight on patient prognosis and allow for tailored therapies. The small sample sizes of included studies means that findings reported in the present review cannot be applied to a broader population with BD. Future studies should evaluate the effects of neuroinflammation in BD and astrocyte function in larger samples. We recommend standardized methodologies, as current studies are limited by the heterogeneity of samples included with regards to eligibility criteria, concomitant medication use, bipolar subtypes, mood states, and comorbidities.

## 5. Conclusions

There is a small body of evidence suggesting that astrocyte dysfunction may be involved in the pathophysiology of BD. Further in vivo assessment of astrocyte activity across mood states would shed further light on the pathophysiology of BD. Longitudinal measures of astrocyte activity pre- and posttreatment could provide insights into the mechanism of the action of first-line, evidence-based treatments such as lithium. Future research should investigate astrocyte-derived exosomes as peripheral markers of astrocyte dysfunction. This line of work will inform the development of novel neuroprotective therapeutics that target astrocytes for the management of BD.

## Figures and Tables

**Figure 1 brainsci-12-01513-f001:**
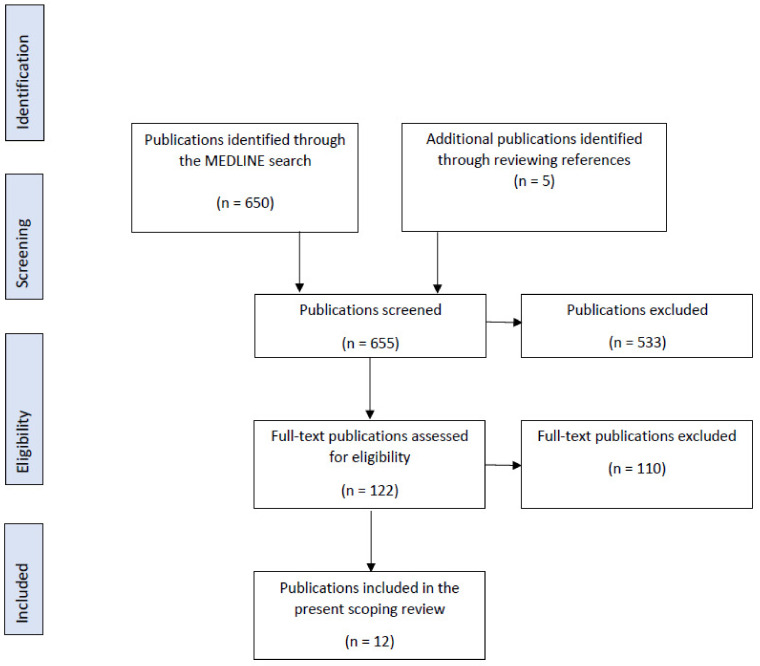
PRISMA Flow Diagram for the scoping review process.

**Table 1 brainsci-12-01513-t001:** Summary of in vitro publications included in the scoping review. Includes publication name, authors, year of publication, type of study, purpose, participant numbers, and main findings.

Publication Name, Authors	Year of Publication	Type of Study	Purpose	Number of Participants (Total/Subjects/Healthy Controls)	Results
Serum S100B and antioxidant enzymes in bipolar patients, Andreazza et al. [24]	2007	In vitro	To evaluate brain injury using serum S100B content and oxidative stress in BD patients.	84/52/32	Significant increment of serum S100B in hypomanic (*p* = 0.004) and manic (*p* = 0.011) and no changes in euthymic patients (*p* = 0.263).
Stem cells, pluripotency and glial cell markers in peripheral blood of bipolar patients on long-term lithium treatment, Ferensztajn-Rochowiak et al. [25]	2018	In vitro	To investigate the effect of long-term lithium treatment on very-small embryonic-like stem cells (VSELs) and the mRNA expression of pluripotency and glial markers in peripheral blood in patients with bipolar disorder (BD).	45/30/15	BD lithium-negative patients exhibited significantly higher numbers of VSELs in comparison to HC.
Elevated serum S100B protein in drug-free bipolar patients during first manic episode: a pilot study, Machado-Vieira et al. [26]	2002	In vitro	To examine the possible effects of mania on S100B turnover in serum.	40/20/20	Statistically significant increased levels of S100B in bipolar mania (Wilcoxon signed-rank test, Z522.45, P50.01).
Decreased S100B serum levels after treatment in bipolar patients in a manic phase, Tsai et al. [27]	2017	In vitro	To investigate the serum levels of S100A10 and brain injury-related biomarkers, including S100B, NSE, and HSP70, in BD patients experiencing a manic phase compared to HC. In addition, to assess the relationship between these markers and Young Mania Rating Scale (YMRS) scores in BD patients during manic episodes and investigate the changes in these markers in BD patients after treatment.	47/17/30	Significantly decreased S100B levels only in bipolar manic patients after treatment (*p* = 0.002), but S100B levels were not significantly different from those in healthy controls (*p* > 0.05).

**Table 2 brainsci-12-01513-t002:** Summary of ex vivo publications included in the scoping review. Includes publication name, authors, year of publication, type of study, purpose, participant numbers, and main findings.

Publication Name, Authors	Year of Publication	Type of Study	Purpose	Number of Participants (Total/Subjects/Healthy Controls	Results
Increased expression of glial fibrillary acidic protein in prefrontal cortex in psychotic illness, Feresten et al. [30]	2013	Ex vivo	To investigate if levels of four astrocyte-specific proteins, i.e., glial fibrillary acidic protein (GFAP), aldehyde dehydrogenase\1 L1 (ALDH1L1), vimentin, and excitatory amino acid transporter 1 (EAAT1), are altered in SCZ and BPD.	104/69/35	High levels of GFAP in SCZ and BD when compared to controls and when comparing individuals with psychotic symptoms against those without.
Evidence for morphological alterations in prefrontal white matter glia in schizophrenia and bipolar disorder, Hercher et al. [31]	2014	Ex vivo	To assess oligodendrocyte, astrocyte, and microglial populations in post-mortem white matter from SCZ, BD, and HC samples.	60/40/20	Significant increase in oligodendrocyte density (*p* = 0.012) and CNPase protein levels (*p* = 0.038) in BD patient groups compared with control samples.
Sex difference in glia gene expression in the dorsolateral prefrontal cortex in bipolar disorder: Relation to psychotic features, Zhang et al. [32]	2020	Ex vivo	To investigate on transcriptional changes of markers of astrocytes, microglia, and oligodendrocytes in BD in relation to suicide, psychotic features, and sex in the DLPFC and ACC.	64/30/34	Significant sex difference of the glia-related gene expression in individuals with BD in the DLPFC. Males compared to females have a higher expression of most detected genes.
Changes in region- and cell type-specific expression patterns of neutral amino acid transporter 1 (ASCT-1) in the anterior cingulate cortex and hippocampus in schizophrenia, bipolar disorder and major depression, Weis et al. [33]	2007	Ex vivo	To characterize the expression pattern and cellular localization of the ASCT-1 protein in two brain regions (ACC and hippocampus) and to determine if this expression pattern is altered in SCZ, BD, and MDD using immunohistochemical techniques.	60/45/15	Significant loss of immunoreactivity on astrocytes, neurons, and interneurons in multiple regions for SCZ and BD patient groups.
Glial fibrillary acidic protein mRNA levels in the cingulate cortex of individuals with depression, bipolar disorder and schizophrenia, Webster et al. [34]	2005	Ex vivo	To evaluate levels of GFAP mRNA for potential reduction in astrocyte density and if it contributes to the decrease in glial density reported in the ACC in major mental illnesses.	60/45/15	Significant decrease in the levels of GFAP mRNA in the white matter underlying the cingulate cortex in subjects with BD and SCZ.
Immunohistochemical Localization of Phosphorylated GlialFibrillary Acidic Protein in the Prefrontal Cortex and Hippocampus from Patients with Schizophrenia, Bipolar Disorder, and Depression, Webster et al. [35]	2001	Ex vivo	To determine the immunohistochemical localization of phosphorylated GFAP (pGFAP) in the PFC and hippocampus and to investigate potential disease-related changes in distribution of pGFAP containing astrocytes.	60/45/15	pGFAP astrocytes found to be adjacent to blood vessels in the white matter of the gyrus and the polymorphic layer of the dentate gyrus. No significant differences between the patient and control groups.
Amygdala astrocyte reduction in subjects with major depressive disorder but not bipolar disorder, Altshuler et al. [36]	2010	Ex vivo	To evaluate the diagnostic tissue samples from subjects with BD, MDD, and SCZ.	44/30/14	No significant differences in neuronal densities were foundbetween patient and HC groups.

**Table 3 brainsci-12-01513-t003:** Summary of combination in vivo and in vitro publications included in the scoping review. Includes publication name, authors, year of publication, type of study, purpose, participant numbers, and main findings.

Publication Name, Authors	Year of Publication	Type of Study	Purpose	Number of Participants (Total/Subjects/Healthy Controls)	Results
Peripheral inflammatory markers associated with brain volume reduction in patients with bipolar I disorder, Tsai et al. [37]	2021	In vivo and in vitro	To investigate plasma levels of sTNF-R1, IL-1β, TGF-β1, MCP-1, YKL-40, and FKN to assess if peripheral inflammatory markers and illness severity may be associated with volume abnormalities in subregions of limbic, frontal, and temporal lobes in BD.	31/16/15	Significant increase of YKL-40 and sTNF-R1 levels associated with lower volumes of the left anterior cingulum, left frontal lobe, right superior temporal gyrus, and supramarginal gyrus.

## Data Availability

Not applicable.

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
