# Peer review of "Astrocytes in the Neuropathology of Bipolar Disorder: Review of Current Evidence"

_brainsci, 2022, doi:10.3390/brainsci12111513_

Round 1

Reviewer 1 Report

The authors provide an interesting overview of astrocytes in the neuropathology of bipolar disorder (BD). Only a few studies were found, although some indicate astrocytic malfunctioning in BD.

Abstract

The abstract could be somehow more informative, i.e., for biomarkers in plasma/serum/blood, which astrocytic-related biomarkers were evaluated, and which were up, down, or non-significant?  The same for ex-vivo/in-vivo.

Introduction

The authors could improve their introduction and include a rationale for the topic chosen – why are they interested in astrocytes and not particularly in other glial cells? What was the motivation to write about astrocytes in BD? Any conflicting evidence regarding another type of glial cells in the field?

Material and methods

Line 115 – Add information if the measurement was in serum or plasma and the length of medication treatment.

Regarding the GFAP study, could you please specify the cell type mRNA was evaluated?

Table one could be organised by type of study (in vivo, ex vivo, in vitro).

Given that most of the studies included in the review refer to the evaluation of single markers (e.g., GFAP, S100B), the quality of the manuscript would improve substantially if the authors could also search for transcriptomic/bulk tissue studies evaluating astrocytic markers, as these would be a lot more informative than single markers. Articles from Gandal, Toker and any other transcriptomic studies available would be substantial to include.

Discussion

The authors could expand their discussion in terms of perspectives to the field. Perhaps include a table with potential markers of interest to be evaluated in the peripheral blood and the current state of the art of the evaluation of astrocytes by PET scan.

The suggestion to examine exomes comes suddenly in the text. The authors could expand more on this.

There is no discussion about the interaction between astrocytes with immune cells, for example, T cells. There are many interesting research on this interaction published recently, for example in schizophrenia, that could be added to the discussion.

Author Response

Comments from Reviewer # 1

The authors provide an interesting overview of astrocytes in the neuropathology of bipolar disorder (BD). Only a few studies were found, although some indicate astrocytic malfunctioning in BD.

  • The abstract could be somehow more informative, i.e., for biomarkers in plasma/serum/blood, which astrocytic-related biomarkers were evaluated, and which were up, down, or non-significant? The same for ex-vivo/in-vivo
    • As suggested we have added more information regarding the included studies in the revised Abstract

  • The authors could improve their introduction and include a rationale for the topic chosen – why are they interested in astrocytes and not particularly in other glial cells? What was the motivation to write about astrocytes in BD? Any conflicting evidence regarding another type of glial cells in the field?
    • As suggested, we have added a rationale for why astrocytes were selected for this scoping review in the revised Introduction.

  • Line 115 – Add information if the measurement was in serum or plasma and the length of medication treatment.
    • As suggested, we have added that the measurement was in serum and the length of medication treatment was 27.7 ± 4.8 days.

  • Regarding the GFAP study, could you please specify the cell type mRNA was evaluated?
    • We could not specify the cell type mRNA was evaluated as this detail was not provided in the published study. Instead, we have indicated the mRNA were retrieved from venous blood samples.

  • Table one could be organised by type of study (in vivo, ex vivo, in vitro).
    • We have organized Table 1 by type of study as suggested by the reviewer.

  • Given that most of the studies included in the review refer to the evaluation of single markers (e.g., GFAP, S100B), the quality of the manuscript would improve substantially if the authors could also search for transcriptomic/bulk tissue studies evaluating astrocytic markers, as these would be a lot more informative than single markers. Articles from Gandal, Toker and any other transcriptomic studies available would be substantial to include.
    • As suggested, we included text regarding transcriptomic/bulk tissue studies that evaluate S100B and another biomarker of interest.

  • The authors could expand their discussion in terms of perspectives to the field. Perhaps include a table with potential markers of interest to be evaluated in the peripheral blood and the current state of the art of the evaluation of astrocytes by PET scan.
    • As suggested, we have added more content to the Discussion on potential biomarkers of interest, including PET radiotracers.

  • The suggestion to examine exosomes comes suddenly in the text. The authors could expand more on this.
    • We have provided further context on why evaluation of exosomes could be an area of interest in the field.

  • There is no discussion about the interaction between astrocytes with immune cells, for example, T cells. There are many interesting research on this interaction published recently, for example in schizophrenia, that could be added to the discussion.
    • Many thanks for this suggestion. As recommended, we have included a paragraph summarizing work in schizophrenia that reports associations between immune cells and astrocytes.

Reviewer 2 Report

This is a very interesting MS about role astrocytes in the neuropathology of bipolar disorderbipolar disorder. I think that after minor revision the MS will provide important contribution to the field and I will certainly cite it in future.

 I was surprised that authors ignored latest advances in the field. There is evidence that bipolar disorder is caused by environmental mismatch. It seems that neuroinflammation caused modern lifestyle desyncronize circadian rhythm that leads to mood changes.  I recommend to authors to read: Rantala MJ., Luoto, S. Borraz-Leon, J. & Krams, I. (2021) Bipolar disorder: An evolutionary psychoneuroimmunological approach Neuroscience and biobehavioral reviews 122, 28-37.

 I think that this psychoneuroimmunological model of bipolar disorder should be presented in the introduction. It would provide a model to be tested.  It would also be fruitful to discuss their findings from this psychoneuroimmunological point of view.  

 Since manic episodes often includes psychotic symptoms, I recommend authors to read and discuss also: Rantala MJ., Luoto, S. Borraz-Leon, J. & Krams, I. (2022) Schizophrenia: the new ethiological synthesis.  Neuroscience and biobehavioral reviews 142, 104894.

 It would provide another explanation why patients with bipolar disorder might have neuroinflammation.

Author Response

Comments from Reviewer # 2

This is a very interesting MS about role astrocytes in the neuropathology of bipolar disorder. I think that after minor revision the MS will provide important contribution to the field, and I will certainly cite it in future.

  • I was surprised that authors ignored latest advances in the field. There is evidence that bipolar disorder is caused by environmental mismatch. It seems that neuroinflammation caused modern lifestyle desynchronize circadian rhythm that leads to mood changes. I recommend to authors to read: Rantala MJ., Luoto, S. Borraz-Leon, J. & Krams, I. (2021) Bipolar disorder: An evolutionary psychoneuroimmunological approach Neuroscience and biobehavioral reviews 122, 28-37.
    • We have added information pertaining to environmental factors affecting bipolar disorder in the revised Introduction.
  • I think that this psychoneuroimmunological model of bipolar disorder should be presented in the introduction. It would provide a model to be tested.  It would also be fruitful to discuss their findings from this psychoneuroimmunological point of view. 
    • As suggested, we have added further content on the psychoneuroimmunological model of bipolar disorder in the revised Introduction.
  • Since manic episodes often includes psychotic symptoms, I recommend authors to read and discuss also: Rantala MJ., Luoto, S. Borraz-Leon, J. & Krams, I. (2022) Schizophrenia: the new etiological synthesis.  Neuroscience and biobehavioral reviews 142, 104894. It would provide another explanation why patients with bipolar disorder might have neuroinflammation.

As suggested, we have added content to the revised Discussion indicating a potential pathophysiological overlap between schizophrenia and bipolar disorder and how this may inform further evaluation of astrocytes in bipolar disorder.

Reviewer 3 Report

A very reliable work, which is a synthesis of knowledge about the involvement of astrocytes in the pathophysiology of bipolar disorder. The authors have done a lot of work that deserves attention. I believe that the article is eligible for publication and I have no critical comments.

Author Response

Comments from Reviewer # 3

A very reliable work, which is a synthesis of knowledge about the involvement of astrocytes in the pathophysiology of bipolar disorder. The authors have done a lot of work that deserves attention. I believe that the article is eligible for publication and I have no critical comments.

  • Thank you for the positive comments regarding our work.

Reviewer 4 Report

The paper is an interesting scoping review of the role of astrocytes in bipolar disorder. The topic is relevant, and the manuscript is clear. I have only a few comments for the authors:

- please revise the first sentences of the abstract because they are too fragmented;

- I think the introduction is not so clear about the connections between BD, astrocytes, inflammatory system. Please, revise.

- Have you conducted the search only in one database? This could be a limit even in a scoping review.

- please include the limitation of your study in the text, both from the methods applied and the current literature.

Overall the paper should be considered for publication after a minor revision. 

Author Response

Comments from Reviewer # 4

The paper is an interesting scoping review of the role of astrocytes in bipolar disorder. The topic is relevant, and the manuscript is clear. I have only a few comments for the authors:

  • please revise the first sentences of the abstract because they are too fragmented;
    • As suggested we have revised the Abstract to improve readability.
  • I think the introduction is not so clear about the connections between BD, astrocytes, inflammatory system. Please, revise.
    • As suggested, we have revised the Introduction to provide further context on the associations between neuroinflammation, astrocytes, and the pathophysiology of BD.

  • Have you conducted the search only in one database? This could be a limit even in a scoping review.
    • The search for the present scoping review was only conducted in MEDLINE. We have added this as a limitation of the study.
  • please include the limitation of your study in the text, both from the methods applied and the current literature. Overall the paper should be considered for publication after a minor revision.
    • As suggested, we have added a paragraph on limitations of the current review and the studies included in the revised Discussion.
